# Local Editing of Cross-Surface Mappings with Iterative Least Squares Conformal Maps

Donya Ghafourzadeh*
École de technologie supérieure, Montréal, Canada

Srinivasan Ramachandran†
École de technologie supérieure, Montréal, Canada

Martin de Lasa‡
Autodesk, Toronto, Canada

Tiberiu Popa§
Concordia University,
Montréal, Canada

Eric Paquette¶
École de technologie supérieure,
Montréal, Canada

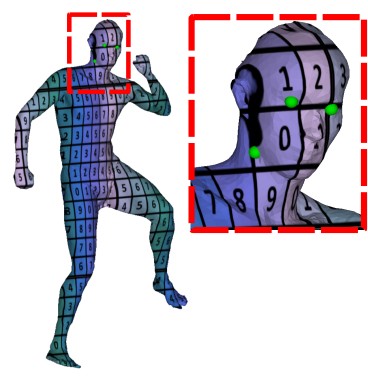 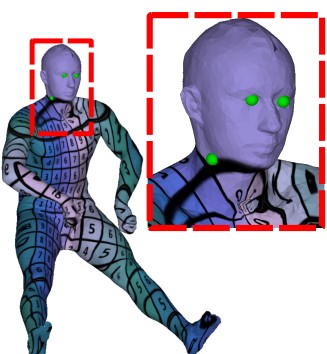 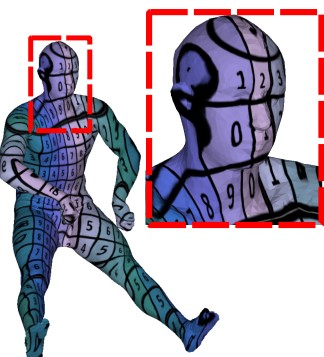

| (a) Mesh $A$ | (b) Mesh $B$, initial mapping [2] | (c) Mesh $B$, edited mapping |

Figure 1: The initial mapping between mesh $A$ and $B$ is globally good, but is locally misaligned in the head region. Using our approach, the mapping is locally improved (c).

## ABSTRACT

In this paper, we propose a novel approach to improve a given surface mapping through local refinement. The approach receives an established mapping between two surfaces and follows four phases: (i) inspection of the mapping and creation of a sparse set of landmarks in mismatching regions; (ii) segmentation with a low-distortion region-growing process based on flattening the segmented parts; (iii) optimization of the deformation of segmented parts to align the landmarks in the planar parameterization domain; and (iv) aggregation of the mappings from segments to update the surface mapping. In addition, we propose a new approach to deform the mesh in order to meet constraints (in our case, the landmark alignment of phase (iii)). We incrementally adjust the cotangent weights for the constraints and apply the deformation in a fashion that guarantees that the deformed mesh will be free of flipped faces and will have low conformal distortion. Our new deformation approach, Iterative Least Squares Conformal Mapping (ILSCM), outperforms other low-distortion deformation methods. The approach is general, and we tested it by improving the mappings from different existing surface mapping methods. We also tested its effectiveness by editing the mappings for a variety of 3D objects.

**Index Terms:** Computing methodologies—Computer graphics—Shape modeling—Mesh geometry models

---

*e-mail: gh.dony@gmail.com

†e-mail: ashok.srinivasan2002@gmail.com

‡e-mail: martin.delasa@gmail.com

§e-mail: tiberiu.popa@concordia.ca

¶e-mail: eric.paquette@etsmtl.ca

## 1 INTRODUCTION

Computing a cross-surface mapping between two surfaces (cross-parameterization) is a fundamental problem in digital geometric processing. A wide range of methods have been developed to find such mappings [6, 17, 21], but no single method results in a perfect mapping in every case. Quite often, the mapping results may be good overall, but some specific, sometimes subtle, semantic features, such as articulations and facial features, may remain misaligned, as illustrated in Fig. 1. These imperfections of the final result are often unacceptable in a production setting where the artist needs a high degree of control over the final result, and will often sacrifice automation of a method for higher control. Typically, improving results using surface mapping methods requires the user to iteratively insert some landmarks and solve for the mapping globally. However, since the imperfections are typically localized to a specific region, a local solution that does not change the mapping globally would be preferred in order to ensure that the method does not introduce artifacts elsewhere on the map.

This paper proposes a surface mapping editing approach providing local and precise control over the map adjustments. The process begins with the inspection of an existing vertex-to-point surface mapping between two meshes. In regions where the mapping exhibits some discrepancy, the user sets landmarks positioned at corresponding locations on both meshes. For each such region, we extract a patch on both meshes in order to localize the changes in the mapping, and we flatten them on a common planar domain. The mapping is improved based on a 2D deformation optimization that steers the landmarks toward correspondence while limiting distortion and having theoretical guarantees to maintain the local injectivity of the map. We developed a new 2D deformation approach denoted Iterative Least Squares Conformal Maps (ILSCM), which iteratively minimizes a conformal energy, each iteration ensuring that flips do not occur, and in practice, ensuring progress toward satisfying the constraints. We chose to work with a conformal energy as we want

to be able to improve mappings where the deformation between the pair of meshes is not isometric. Our editing approach can successfully align the mapping around landmarks without any degradation of the overall mapping. The local surface maps are extracted from their respective deformed segments and parameterization domains, and are then combined to form an improved global surface mapping.

Our approach solves an important practical problem and offers three novel scientific contributions. The first is a practical approach for local surface map editing, which we show, using both qualitative and quantitative metrics, provides better results than other state-of-the-art methods. The second involves a compact segmentation which results in a compromise between a low-distortion flattening and a low-distortion deformation when aligning the landmarks. The third is a new deformation approach, ILSCM, which preserves conformal energy better than other state-of-the-art methods, and that has theoretical guarantees preventing the introduction of fold-overs.

## 2 RELATED WORK

While a lot of research has been done on creating correspondences between 3D objects, comparatively fewer methods have been proposed on correspondence editing. Nguyen et al. [16] measure and optimize the consistency of sets of maps between pairs belonging to collections of surfaces. They compute a score for the map, and then apply an optimization to iteratively improve the consistency. The limitation of their method lies in its requirement of having multiple maps instead of a single map for a pair of surfaces. Ovsjanikov et al. [18] propose the functional maps representation to establish correspondences between surfaces based on Laplacian eigenfunctions rather than points. Working in that smooth basis function space makes it easy and efficient to generate smooth mappings, but significant modifications to the underlying method would be required to allow local adjustments of the mapping guided by the user. Another limitation is that their method is limited to near-isometric surfaces since non-isometric deformation overrides the assumption that the change of basis matrix is *sparse*. Ezuz and Ben-Chen [9] remove the isometric restriction in their proposed method for the deblurring and denoising of functional maps. They smooth a reconstructed map by mapping the eigenfunctions of the Laplacian of the target surface in the span of the source eigenfunctions. Their technique can be incorporated into existing functional mapping methods, but selecting the right number of eigenfunctions to perform the denoising is difficult. Compared with a ground truth mapping, increasing the number of eigenfunctions decreases the error until a minimum is reached, but adding more eigenfunctions beyond this point increases the error. While this can be observed on a ground truth mapping, there are no methods to achieve the minimum error for an arbitrary mapping. Gehre et al. [10] incorporate curve constraints into the functional map optimization to update the mapping between non-isometric surfaces. They provide an interactive process by proposing a numerical method which optimizes the map with an immediate feedback. While their method is not limited to isometric surfaces, it does however need several curve constraints to obtain a meaningful functional map.

Vestner et al. [28] improve dense mappings even in the case of non-isometric deformations. Their method is an iterative filtering scheme based on the use of geodesic Gaussian kernels. An important restriction of their method is that it requires both surfaces to be discretized with the same number of vertices and vertex densities. Panozzo et al. [19] propose the weighted averages (WA) on surfaces framework. They use WA with landmarks in order to define mappings and then the user can improve the mapping by adjusting the landmarks. Although WA generates good mapping results, its improved mapping application cannot use a mapping as input. Since most state-of-the-art methods improve the mapping globally, this makes it hard for the user to fine-tune the mapping without risking modifying areas that should not be affected. Furthermore, some

methods face significant limitations such as being constrained to isometric deformations and requiring compatible meshes on both surfaces.

An alternative way to frame the correspondence editing problem is as a deformation method in a planar parameterization space. Least Squares Conformal Maps (LSCM) [14] apply a deformation energy which contains a term for preserving surface features and a term for position constraints. In contrast, Jacobson et al. [12] provide a smooth and shape-preserving deformation using biharmonic weights. The main drawback of the LSCM and biharmonic weights methods is that they can introduce fold-overs while deforming the mesh. Injectivity is a key property we want to achieve in our mapping editing, but extracting a mapping from meshes with fold-overs breaks this property. A strategy to provide locally injective maps [8, 15] is to apply inequality constraints solely to the boundaries. First, an initial locally injective map is generated using a parameterization algorithm and then it is optimized when adhering to a specific distortion bound. The main problem is that this solution has suboptimal distortion. A recent series of papers [20, 23, 25] propose a strategy where the energy of a triangle tends to infinity as the triangles become degenerate, and thus, any locally minimal solution will be, by construction, exempt of fold-overs. While this is an elegant method, the problem is that in some cases where, due to the user constraints, some triangles will come close to becoming degenerate, they carry a disproportionately high share of the total energy as compared to the rest of the triangles. For our mapping editing application, such cases occur frequently, and we observed (Sec. 3.3) that in their presence, Locally Injective Mappings (LIM) [23] and Scalable Locally Injective Mappings (SLIM) [20] often produce an inferior result both qualitatively and quantitatively. Golla et al. [11] outperform LIM and SLIM by modifying the Newton iteration for the optimization of nonlinear energies on triangle meshes. They analytically project the per-element Hessians to positive semidefinite matrices for efficient Newton iteration and apply global scaling to the initialization.

In this work, we propose to edit a surface mapping by locally adjusting the mapping to align landmarks set by the user. To move the landmarks toward their expected positions, we deform a local segmented patch of the mesh. We found that current deformation methods had drawbacks (flipped triangles and high distortion) forbidding their use in our mapping editing framework. We thus derived a new deformation approach that iteratively minimizes a conformal energy, making sure that in each iteration we have no flipped triangles. More specifically, our ILSCM approach optimizes the quadratic LSCM energy, but it relaxes the user constraints to avoid flips. Therefore, after each iteration, the user constraints may not be satisfied, but by repeating the process, we reach a configuration that has low conformal energy (lower than LIM or SLIM), and the user constraints are guaranteed to be better satisfied than initially. In practice, the user constraints are always satisfied up to a user-provided epsilon. Our approach opens the door to a new family of deformation methods that optimize the deformation by effectively finding an optimal flow of the vertices based on conformal energy minimization.

## 3 SURFACE MAPPING EDITING

As explained in the introduction, mappings computed by state-of-the-art (automatic) methods are often globally good, but locally wrong in a few areas. We provide an approach to locally improve the surface mapping. The user typically inspects the mapping visually through texture transfer. In local regions where the mapping should be improved, the user sets landmarks at locations that should correspond on the pair of surfaces (Fig. 2a). We edit the mapping by deforming parts of the meshes with respect to each other (Fig. 2c) to improve the alignment of the user-provided landmarks, and then we rebuild the mapping from the deformed parts (Fig. 2d).

Our main goal is to obtain a low distortion of the meshes at

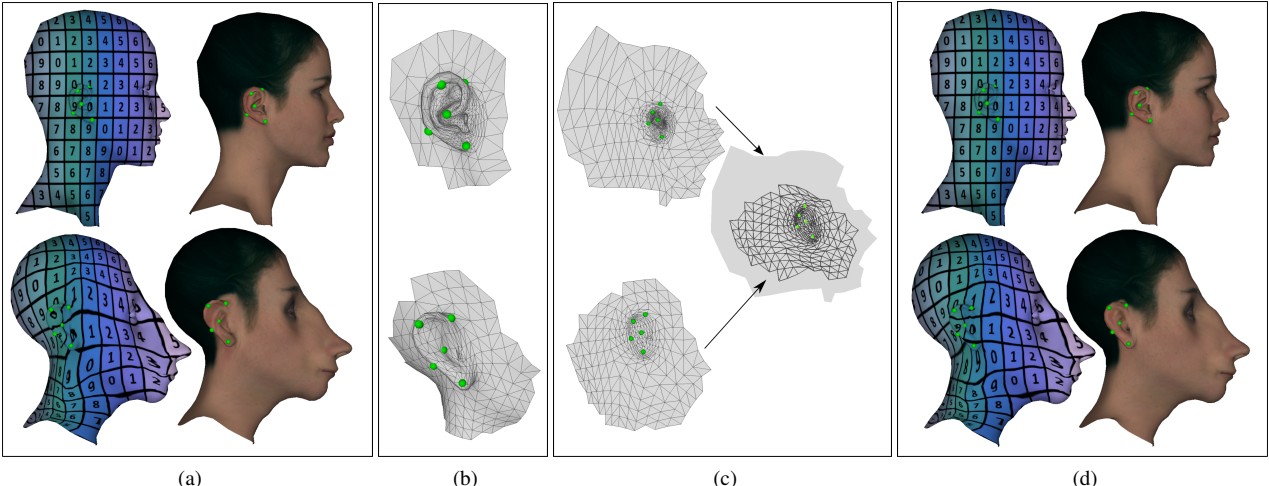

(a)       (b)       (c)       (d)

Figure 2: Overview of the approach: (a) Inspection of the initial surface mapping between meshes $A$ and $B$. In mismatching regions, the user sets landmarks at locations that should correspond (one such region shown); (b) Segmentation for the local adjustment of the surface mapping; (c) Planar parameterization and deformation; (d) Edited surface mapping.

each phase of our approach. To deform the mapping with a good control on distortion, we conduct a planar parameterization of the meshes. Since the planar parameterization of smaller segments of a mesh leads to less distortion versus when computed for the entire mesh, we do a segmentation based on user-identified local *regions* where the mapping needs to be updated. Afterwards, we deform one segment with respect to the other in the planar parameterization space. The deformation is aided by our new ILSCM, ensuring that the deformation causes limited distortion on the meshes. Finally, the mapping is extracted based on how segments overlap each other. Our approach has four key phases:

1. Initial mapping inspection (Sec. 3.1, Fig. 2a): The user inspects the mapping, identifies the mismatching regions of the meshes, and sets landmarks for each region.

2. Segmentation (Sec. 3.2, Fig. 2b): From the mismatching regions, an automatic segmentation is derived.

3. Mapping deformation (Sec. 3.3, Fig. 2c): A localized deformation improves the alignment of the landmarks and the smoothness of the mapping.

4. Mapping extraction (Sec. 3.4, Fig. 2d): Partial mappings are extracted from the pairwise parameterization spaces of the segments, and aggregated to locally update the mapping.

### 3.1 Initial Mapping Inspection

Our approach works with input meshes $A$ and $B$, along with a vertex-to-point surface mapping between the meshes. The mapping links each vertex of a mesh to a barycentric coordinate on the other mesh. It should be noted that our approach works regardless of the initial method used to establish the surface mapping, as long as a dense mapping is provided.

Given a mapping, the user will visualize it using texture maps to identify mismatching *regions*. These correspond to isolated zones of the meshes where the mapping is incorrect. For each region, the user sets corresponding landmarks on both meshes at locations that should match each other. The landmarks for each region $i$, $L_A(i) = \{l_A(i)_1, l_A(i)_2, \ldots, l_A(i)_k\}$ and $L_B(i) = \{l_B(i)_1, l_B(i)_2, \ldots, l_B(i)_k\}$, are expressed as barycentric coordinates on $A$ and $B$, respectively.

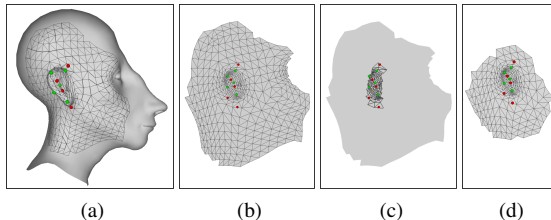

(a)     (b)     (c)     (d)

Figure 3: Steps of the segmentation phase. Step 1: Initial patch, (a) in 3D and (b) after flattening; (c) Step 2: Extraction of the faces touching the convex hull of the landmarks; (d) Step 3: Geodesic growth of the compact patch from the convex hull faces.

### 3.2 Segmentation

The user provides hints where the mapping needs to be modified by setting pairs of landmarks on both meshes. In order to keep the map editing local, a segment is identified on both meshes where the map editing will be performed. Computing such a segment is not trivial as there are a number of requirements: the segment should be a single connected component with disk topology, should be compact, and should contain all the landmarks of the region $i$. The size of the segment is also important. If the segment is too large we may lose locality, but if it is too small, we may introduce further distortion if the vertices need to move over a long distance. We assume that outside these segments, the mapping is satisfactory, and it can be used to set boundary conditions when deforming a segment with respect to the other to align the landmarks.

Our segmentation approach has three steps. In the first step (Fig. 3a), we grow an initial patch on the 3D surface from the landmarks, ensuring that it is one connected component, that it encloses all of the landmarks, as well as the positions corresponding to the landmarks from the other mesh. We flatten this patch in 2D (Fig. 3b), where we have more tools available to control the size and shape of the patch. In the second step, we compute a compact 2D patch from the convex hull of the landmarks in the 2D space (Fig. 3c), and ensure that we fill any artificial internal boundaries (internal "holes" with polygons from the full mesh missing). In the third step, we grow the compact patch from the previous step to allow enough room between the boundary and the landmarks

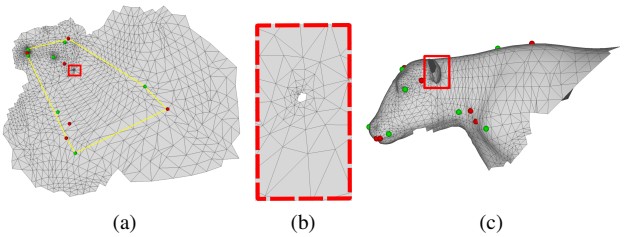

(a)    (b)    (c)

Figure 4: The convex hull filling of step 2 is important to avoid artificial boundaries. (a) Flattened mesh with convex hull. Region highlighted in red contains an artificial internal boundary; (b) Zoom-in and (c) 3D mesh, both showing the artificial boundary and missing faces.

(Fig. 3d), preparing the ground for a low-distortion deformation phase (Sec. 3.3). This segmentation is applied to each region of meshes $A$ and $B$ independently. We now explain in more detail the process as applied to one region $i$ of mesh $A$, but the same is also conducted for mesh $B$ and other regions.

Based on the mapping from mesh $B$ to $A$, corresponding landmark positions are calculated on $A$ for each landmark of $L_B(i)$, yielding $CP_{B \to A}(i) = \{cp_{B \to A}(i)_1, cp_{B \to A}(i)_2, \ldots, cp_{B \to A}(i)_k\}$. The goal of the first step is to extract an *initial patch* and to flatten it. To meet the two conditions of (1) having a single connected component and (2) containing all of the landmarks from $L_A(i)$ and $CP_{B \to A}(i)$, we will compute the union of face groups computed by identifying faces around each landmark $l_A(i)_j$ and around each corresponding position $cp_{B \to A}(i)_j$. For each, starting with the face containing the point, we iteratively add rings of faces until the group of faces contains at least half plus one of the landmarks from $L_A(i)$ and $CP_{B \to A}(i)$. The requirement to include half plus one of the landmarks ensures that when we combine the groups of faces, this initial patch meets the two conditions. This procedure results in a "disk with holes" topology, which is sufficient to flatten the patch using ABF++ [24].

One disadvantage of the initial patch is that it can contain concavities, and even internal "holes". While concavities are not a problem for the flattening, they provide poor boundary conditions, making it harder to smoothly distribute the deformation error. From the initial patch of step one, the second step extracts a *compact patch* that surrounds the landmarks. To this end, we identify the convex hull of the landmarks in the 2D parameterization space. Then, we only consider the faces which have at least one of their vertices within the convex hull (faces identified in black in Fig. 3c). The use of the convex hull results in a patch exempt of large concavities in its boundary. Nevertheless, depending on the meshes and the arrangement of landmarks, some of the initial patches have "holes" with polygons from the full mesh missing, creating artificial internal boundaries in the patch (Fig. 4). We add the missing faces by analyzing the inner boundaries (holes). Filling the whole by adding the missing faces from the full mesh has the advantage of preventing unwanted deformation that would result from such artificial boundaries, and it ensures that there are no internal areas within the region where the mapping would not be adjusted.

The third step tries to balance the conflicting goals of having a small versus a large patch. As can be seen in Fig. 5, the larger the patch, the greater the stretch distortion [22] between the patch in 3D and after flattening to 2D. The distortion would be even higher if flattening the whole mesh (Fig. 6). Conversely, a smaller patch means that the landmarks are closer to the boundary, and the deformation that aligns the landmarks will induce more distortion to the triangles between the boundary and the landmarks. We thus want to grow the compact patch to ensure that there is enough room around each landmark and corresponding landmark position pairs to diffuse the distortion from the deformation phase. We also know that as we

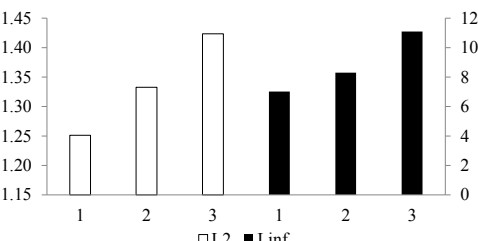

Figure 5: $L_2$ and $L_{inf}$ graphs showing the average stretch distortion [22] of six patches between their configuration in 3D and after flattening by ABF++ [24]. The labels 1, 2, and 3 correspond to patch sizes in ascending order. Smaller patches result in lower distortion, and thus, a configuration more faithful to the 3D mesh.

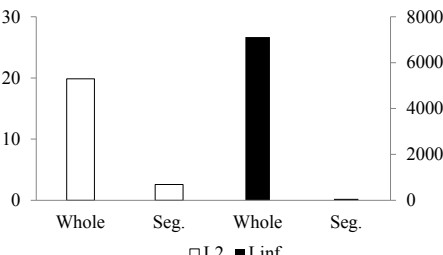

Figure 6: Flattening the segmented patch (Seg.) results in significantly lower $L_2$ and $L_{inf}$ stretch distortion [22] as compared to flattening the whole mesh. Distortion is calculated between the surface in 3D and after flattening.

get closer to the landmarks, we are getting closer to the areas where the mapping is wrong, and as such, extending outwards is necessary in order to have a good boundary condition for the deformation. Regarding how far away the patch should be extended, we use a distance proportional to the geodesic distance (on the 3D mesh) between the landmark and its corresponding landmark position, adding faces that are within that distance from each landmark of the pair. We compared the distortion between the 3D patch and the 2D patch for 1 to 10 times the geodesic distance. Fig. 7 shows a pattern where very small patches do not have enough room to move the landmarks without high distortion. At the same time, patches that are too large also exhibit large distortion because of the flattening from 3D to 2D. As can be seen in Fig. 7, a good compromise between the two sources of distortion is around two times the geodesic distance. Accordingly, we add triangles around each landmark until we reach a geodesic distance bounded to two times the geodesic distance (on the 3D mesh) between the landmark and its corresponding landmark position.

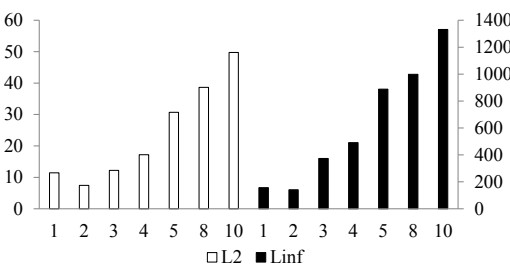

Figure 7: $L_2$ and $L_{inf}$ graphs showing the average distortion with respect to different geodesic distances used in step three. The graph shows the average distortion of six patches between their configurations in 3D and 2D after deformation (Sec. 3.3).

It is necessary to apply steps one and two, because step three alone could lead to disconnected components or artificial internal boundaries that do not exist in the mesh, but that exist in the patch because of the patch growth process (Fig. 4). The sequence of steps one to three provides the final segments from $A$, $\{A_1, A_2, \ldots, A_n\}$, and the same for mesh $B$, yielding $\{B_1, B_2, \ldots, B_n\}$. These final segments are flattened through ABF++ and we refer to them as $\left\{\widetilde{A_1}, \widetilde{A_2}, \ldots, \widetilde{A_n}\right\}$ and $\left\{\widetilde{B_1}, \widetilde{B_2}, \ldots, \widetilde{B_n}\right\}$. We selected ABF++ to flatten our segments as we can assume it will yield a parameterization exempt of flipped triangles (injective).

### 3.3 Mapping Deformation

Our main observation is that the initial surface mapping is globally adequate, but wrong in localized regions. With this assumption, we line up the boundary of the regions by relying on the surface mapping. We will then deform the interior to align the landmarks while keeping a low distortion.

As we have two segmented meshes, there are two ways to align the landmarks: deform $\widetilde{A}_i$ on $\widetilde{B}_i$ or $\widetilde{B}_i$ on $\widetilde{A}_i$. We select the one with the lower $L_2$ distortion [22] (between the segment in 3D and 2D), keeping it fixed and deforming the other. Here, we will explain the deformation of $\widetilde{B}_i$ with respect to $\widetilde{A}_i$, but the deformation of $\widetilde{A}_i$ with respect to $\widetilde{B}_i$ proceeds in the same way. We deform $\widetilde{B}_i$ in order to implicitly adjust the mapping by applying an energy minimization. This is achieved by using positional user constraints ($E_L$ – aligning the landmarks, $E_B$ – aligning the boundary) coupled with a distortion preventing regularization ($E_D$ – globally deforming $\widetilde{B}_i$ with low distortion), leading to the following equation:

$$E(V) = E_L(V) + E_B(V) + E_D(V). \tag{1}$$

The user constraints are enforced by soft constraints as follows:

$$E_L(V) = \lambda \sum_{j=1}^{k(i)} \left|\left| l_{aj} - (v_{(j,1)}\beta_{(j,1)} + v_{(j,2)}\beta_{(j,2)} + v_{(j,3)}\beta_{(j,3)}) \right|\right|^2 \tag{2a}$$

$$E_B(V) = \lambda \sum_{j \in \Omega(\widetilde{B}_i)} \left|\left| v_j - \text{map}(v_j) \right|\right|^2, \tag{2b}$$

where $k(i)$ is the number of landmarks of segment $i$; $l_{aj}$ are the landmarks on $\widetilde{A}_i$; vertices $v_{(j,1)}$, $v_{(j,2)}$, and $v_{(j,3)}$ correspond to the three vertices of the triangle on $\widetilde{B}_i$ containing the related landmark $l_{bj}$; and $\beta_{(j,1)}$, $\beta_{(j,2)}$, and $\beta_{(j,3)}$ are the barycentric coordinates. We use $\Omega(\widetilde{B}_i)$ to denote the set of vertices on the boundary of $\widetilde{B}_i$, and $\text{map}(v_j)$ to denote the corresponding position of $v_j$ on $\widetilde{A}_i$ based on the mapping. The energy $E_B$ pulls the vertices of the boundary of $\widetilde{B}_i$ to the positions on $\widetilde{A}_i$ where they correspond given the mapping. When $\lambda$ is small, the map will be injective, but the constraints are generally not satisfied (ultimately, if $\lambda = 0$, $\widetilde{B}_i$ stays the same). Conversely, when $\lambda$ is large, the user constraints are satisfied, but flips may be introduced ($\lambda = \infty$ corresponds to using hard constraints).

#### 3.3.1 Deformation Energy

An ideal deformation energy $E_D(V)$ must meet three criteria: preserving the shape, maintaining the injectivity of the mapping (i.e., no flipped triangles), and satisfying the user constraints as much as possible. For shape preservation, we experimented with several $E_D$: LSCM [14], LIM [23], SLIM [20], and KP-Newton [11]. Each of the energies has a number of pros and cons. LSCM preserves the shape the best, but tends to introduce flips, as illustrated in Fig. 8c. LIM, SLIM, and KP-Newton on the other hand, guarantee injectivity (no flips), but introduce more distortion (between $\widetilde{B}_i$ before and after

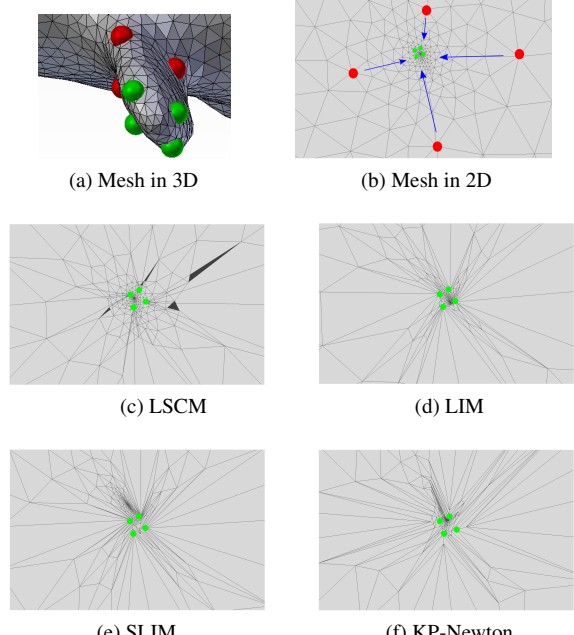

(a) Mesh in 3D      (b) Mesh in 2D

(c) LSCM      (d) LIM

(e) SLIM      (f) KP-Newton

Figure 8: This is an example where we want to adjust the mapping by moving the red landmarks to the positions of the corresponding green landmarks. All deformations were done with the same patch boundary (the one from the ABF parameterization) and they all minimize the same LSCM energy.

deformation) than LSCM. The graph in Fig. 9 illustrates these observations: LSCM has the least distortion, but flipped triangles would destroy the injectivity of the mapping. LIM, SLIM, and KP-Newton have no flips, but overall, they have more distortion as compared to LSCM. LIM minimizes a joint energy where one term optimizes the distortion and the second term optimizes the flips. In such joint optimization frameworks, no one term may be close to a minimum of its own, as shown in Fig. 9, where the results from LIM are worse in terms of the distortion energy than LSCM.

Since all these methods have shortcomings, we propose an approach that bridges the gap between the shape preservation of the original LSCM formulation with the injectivity preservation of LIM, SLIM, and KP-Newton. Our approach, Iterative LSCM (ILSCM), is a different approach where we iteratively optimize to decrease $E(V)$, while preventing flips from occurring. ILSCM performs iterative LSCM steps. The first iteration uses the cotangent weights from segment $\widetilde{B}_i$. The deformed segment from the first iteration is then used to set the weights for the second iteration, and so on. At each iteration, if a triangle flip is detected, we decrease the value of $\lambda$ and redo the same iteration. This way, we are guaranteed to eventually find a $\lambda$ that prevents flips from occurring.

We will now explain how we adaptively adjust $\lambda$ to guarantee that we have no flips, while making as much progress as possible toward achieving the user constraints. In order to measure if the constraints are satisfied, we consider the *initial* maximal distance between any landmark and corresponding landmark position pair $\text{dist}_0 = \max_j \left\{ \left|\left| l_A(i)_j - cp_{B \to A}(i)_j \right|\right| \right\}$, and iterate until the current maximal distance is below the threshold $\varepsilon = \text{dist}_0 / 250$. Appendix A demonstrates that since the progression of landmarks is continuous with respect to $\lambda$, the approach will always find a $\lambda$ that prevents having any flips and that enables progress toward the user constraints. The progress could asymptotically stop, but in all cases, we are guaranteed to prevent triangles from flipping and we limit the

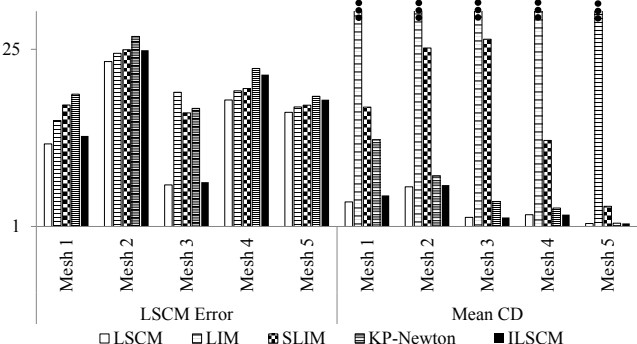

Figure 9: The graph compares results in terms of the residual error of LSCM energy [14] and Conformal Distortion [15] for the deformation of five different 2D meshes ( Fig. 2 and 8, as well as the gingerbread man, dragon, and disk examples from the accompanying video).

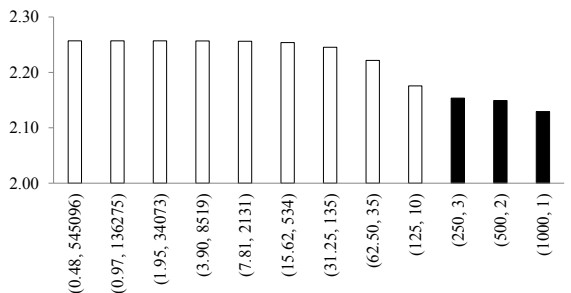

Figure 10: This graph compares results in terms of the residual error of LSCM energy [14] for fixed values of $\lambda$. The horizontal axis labels present $\lambda$ and the number of iterations ($\lambda$, #iter). The black bars highlight results with flipped triangles.

mesh distortion. For the mapping adjustment application, all of the examples we tested converged to meet the user constraints.

A larger $\lambda$ will converge faster, but increases the likelihood of flipped triangles (Fig. 10, black bars). A small $\lambda$ decreases the probability of flipped triangles, but increases the number of iterations needed to satisfy the user constraints. As can be seen in Fig. 10, whether using a small or large $\lambda$, the conformal residual is almost the same and it plateaus for smaller values of $\lambda$. Consequently, even in the theoretical case where our approach would take very small incremental steps, the solution remains valid in the sense that it meets the constraints and the conformal residual remains close to the solution with larger steps. In our experiments, we start with $\lambda = 1000$. After each iteration, we automatically detect if there are flipped triangles, and if so, we redo the deformation of the iteration with $\lambda = \lambda/2$. Fig. 11a demonstrates that the movement of a landmark is continuous with respect to different values of $\lambda$. Fig. 11b further shows that even with small values of $\lambda$, we make progress toward satisfying the constraints.

### 3.4 Mapping Extraction

As the last phase of our approach, we update the surface mapping between $A$ and $B$ from the planar parameterizations. We first extract the mappings from each pair $(\widetilde{B}_i, \widetilde{A}_i)$. Then, we aggregate and transfer them to $A$ and $B$. With the mapping being expressed as barycentric coordinates on the other mesh, we can update it by simply getting the barycentric coordinates of vertices from $\widetilde{B}_i$ to faces of $\widetilde{A}_i$ and vice-versa.

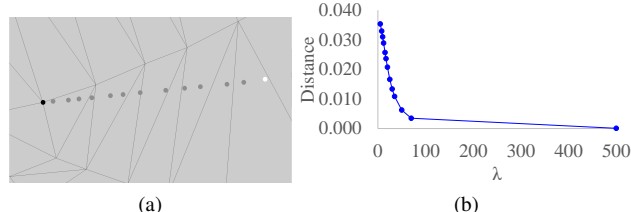

| (a) | (b) |

Figure 11: This figure shows the impact of $\lambda$ for an example with a single landmark. (a) The black point is the initial position of the landmark, the white point is the constraint position, and the grey points show landmark positions corresponding to different values of $\lambda$ (computing a single iteration from the initial configuration for each). We can see that the different positions of the landmark are very continuous and predictable. (b) For the same positions shown in (a), we plot the values of the distance against the value of $\lambda$. We see that we make progress toward satisfying the user constraints even for small values of $\lambda$. We also see that it is to our advantage to begin with a large value of $\lambda$ to reduce the number of iterations before convergence.

## 4 RESULTS

We validate our approach with various cases of faces, as well as with a wider range of objects with different morphologies from different data sets and artist contributions (see Table 1). Experiments are presented based on different initial mapping methods: orbifold tutte embeddings [1], Elastiface [6], deformation transfer [26], functional mapping [17], weighted averages [19], and joint planar parameterization [21]. The number of landmarks and segments is proportional to the quality of the initial surface mapping and the complexity of the objects (see Table 1). We evaluate the capabilities of our approach based on a qualitative evaluation by visual inspection and a quantitative evaluation based on geodesic distance.

### 4.1 Qualitative Evaluation

Our approach prevents flipped triangles, and, essentially, it preserves the shape, while satisfying the user constraints. Furthermore, it distributes the deformation error more uniformly across the mesh surface. As can be seen in Fig. 9, our distortion energy is lower than SLIM, is often lower than LIM, and is only slightly greater than LSCM. We believe that this optimization strategy is more suited to this type of problem than a joint optimization strategy. Fig. 12 compares LSCM, LIM, SLIM, and KP-Newton to our ILSCM (iterated 257 times and final $\lambda$ was 31.25) for the example from Fig. 8. ILSCM distributed errors more uniformly over the whole deformed mesh, as compared to LIM, SLIM, and KP-Newton. The accompanying video shows how our iterative approach progressively conducts the deformation, in comparison to LSCM, LIM, SLIM, and KP-Newton. The meshes we deform in the video are the same as some of the examples from the LIM paper [23]. For a fair comparison, we perform SLIM, LIM, and KP-Newton, all using the LSCM energy [14] for all the examples in the paper as well as the video.

For LIM, we apply a 1E−12 barrier weight, which is sufficient to prevent flips. We experimented with barrier weights of LIM ranging from 1E−4 to 1E−20. Barrier weights smaller than 1E−12 had an imperceptible impact, while those equal to or lower than 1E−20 did not converge. For each deformation energy, we experimented with two different initial states: weights from the 3D triangles and weights from the flattened $\widetilde{B}_i$. The distortion between $\widetilde{B}_i$ before and after deformation was lowest when deforming using the weights from the flattened $\widetilde{B}_i$. We thus used the weights from the flattened $\widetilde{B}_i$.

Visual inspection of results is a common form of validation for

Table 1: Information about the meshes, data sets, number of landmarks, segments, and computation time.

| Meshes | Mapping Method | Data set | Segments | Mesh # Faces | Seg. # Faces | # Land-marks | $\lambda$ | # Iter. | Seg. Time (s) | Def. Time (s) |
|---|---|---|---|---|---|---|---|---|---|---|
| Fig. 20: Fish | [21] | SHREC07 [27] | Fins | 8-20K | 1K, 1K | 2, 3 | 31.25, 62.5 | 128, 101 | 3.23 | 4.64 |
| Fig. 20: Aircrafts | [21] | SHREC07 [27] | Stabilizer | 11K | 3K | 4 | 125 | 40 | 4.13 | 1.30 |
| Fig. 13: Man | [2] | FAUST [5] | Head | 13K | 5K | 4 | 1000 | 1 | 2.37 | 1.22 |
| Fig. 15: Ilana-Eagle | [26] | Character Generator | Neck, Nose | 6-10K | 1-2K | 3, 7 | 1000, 1000 | 1, 1 | 4.55 | 1.12 |
| Fig. 16: Oldman-Samburu | [26] | Character Generator | Ear | 6-11K | 2K | 5 | 125 | 121 | 1.94 | 3.61 |
| Fig. 1 and 18: Man | [2] | SCAPE [3] | Head | 24K | 11K | 4 | 1000 | 27 | 2.91 | 8.67 |
| Fig. 19: Horse-Cow | [17] | SHREC07 [27] | Head | 8-11K | 7K | 7 | 125 | 6 | 4.14 | 1.27 |
| Fig. 19: Lamb-Dog | [17] | SHREC07 [27] | Head | 3-7K | 1K | 6 | 62.5 | 31 | 1.36 | 0.84 |
| Fig. 22: Tiger Woman | [19] | Character Generator | Nose | 3K | 1K | 2 | 500 | 2 | 0.22 | 0.16 |
| Fig. 21: Bird | [2] | SHREC07 [27] | Tail | 10-15K | 3K | 4 | 250 | 46 | 1.48 | 1.50 |
| Fig. 21: Wolf | [2] | TOSCA [7] | Head | 8K | 3K | 4 | 1000 | 6 | 2.09 | 1.27 |
| Fig. 23: Ilana-Badger | [26] | Character Generator | Ear, Nose, Mouth | 1-6K | 1K, 1K, 1K | 5, 8, 7 | 1000, 125, 1000 | 1, 6, 1 | 4.76 | 0.99 |
| Fig. 24: Man-Curve | [29] | Artist and Character Generator | Thumb | 2-8K | 854 | 2 | 1000 | 1 | 1.35 | 0.20 |

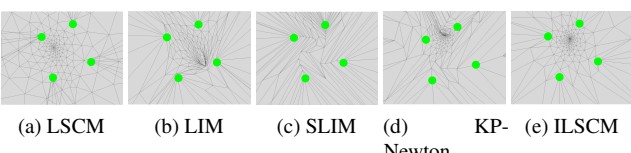

(a) LSCM    (b) LIM    (c) SLIM    (d) KP-Newton    (e) ILSCM

Figure 12: Different methods all minimizing the LSCM energy. ILSCM does not generate any flips, and additionally, it better preserves the shape of the triangles compared to LIM, SLIM, and KP-Newton.

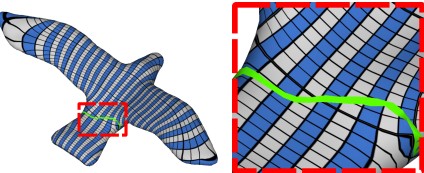

Figure 14: Figure showing that our edited mappings are smooth across the boundary of the edited regions.

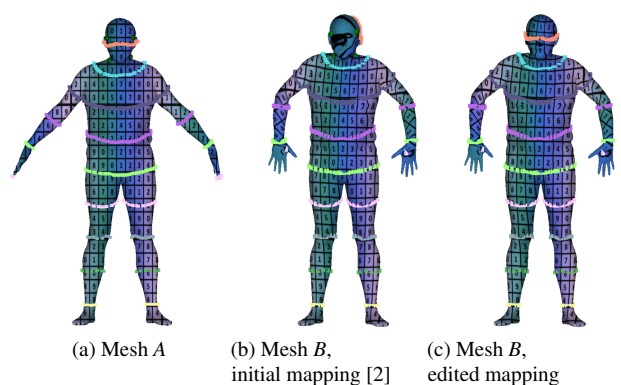

(a) Mesh $A$    (b) Mesh $B$, initial mapping [2]    (c) Mesh $B$, edited mapping

Figure 13: Isopoints and grid texture visualization showing the local improvement of the initial mapping in the head region.

mapping problems. We use a visualization method based on texture transfer. We copy texture coordinates from one mesh to the other using the mapping, setting the $uv$ coordinates of a vertex to the barycentric interpolation of the $uv$ coordinates of the other mesh. For this visualization, we used two different types of textures. The first type was a grid texture. Fig. 1, 2, 13, 15, 16, as well as Fig. 19-23 qualitatively show that we obtain considerably better mappings using our editing approach.

An important assumption of our approach is that we can edit the mapping locally. This implies that it is important to have a smooth transition at the boundary of the regions where we conduct local editing. Fig. 14 shows a typical example of the smoothness of our edited mapping across the boundary of the segmented region. The accompanying video also compares the transition of the mapping

across the boundary by transferring texture from mesh $A$ to mesh $B$ using both initial mapping and edited mapping for the test cases of Fig. 13, Fig. 20 (top row), and Fig. 21.

For the specific case of faces, we use realistic facial textures, making it easier to highlight important semantic facial features. These features are derived from three important considerations: modeling, texturing, and animation. A realistic facial texture is often enough to highlight modeling and texturing issues. Problems around the nose (Fig. 15 and 23), lips (Fig. 15 and 23), and ears (Fig. 16 and 23) are easy to spot with such a texture visualization approach. These examples show cases that are ideal for our approach: the initial mappings are globally good, with few local misalignments. Instead of solving for the mapping globally, our approach provides a local solution for these specific semantic regions. For facial animation, other features need to be identified in the textures. Accordingly, some of our texture visualizations use a set of curves that are positioned relative to the areas that deform during animation, based on the facial anatomy. Fig. 15 illustrates the improvement in the correspondence of these animation-related features as compared against the initial surface mapping.

Our approach assumes that the segments can be flattened to 2D without any flipped triangles. While the hypothesis is essential to get injective mappings, our approach is still robust to cases where the flattened segments would contain flipped faces. Meshes used in the industry often exhibit small discrepancies such as cracks, holes, and handles. Fig. 16 presents such a case where one of the meshes is of a different genera (contains two handles in the ear region). Although it is not possible to get injective mappings when dealing with different genera, our approach behaves robustly: it can improve the mapping in the region with different genera and it does not degrade the mapping in the edited region nor in its vicinity. Furthermore, even if it is not possible to achieve injective mappings in such cases, our edited mappings have reasonable properties: the mapping from the lower genera ($A \rightarrow B$) is injective, and the mapping

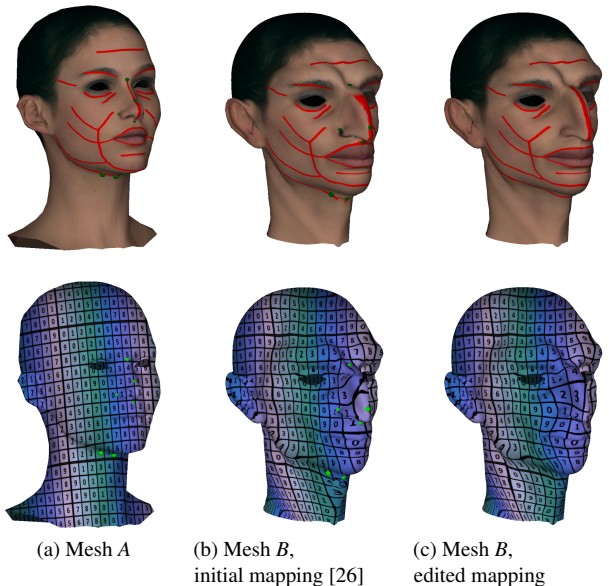

(a) Mesh *A*  (b) Mesh *B*, initial mapping [26]  (c) Mesh *B*, edited mapping

Figure 15: Map editing in the facial area. Realistic facial textures and grid textures are used for qualitative evaluation. Improvement of the initial surface mapping in the nose, mouth, and neck regions is apparent through improved red curves of (c) top.

from the higher genera ($B \rightarrow A$) is surjective.

## 4.2 Quantitative Evaluation

While the qualitative evaluations of Sec. 4.1 demonstrate that our approach results in clear improvements, we also quantitatively measure how our approach improves the mappings. We first use the same process as in the paper of Kim et al. [13] in order to measure the accuracy of the surface mapping. Their method transfers vertex positions to the other mesh using the mapping under evaluation and a ground truth mapping. It then computes the geodesic distances from the corresponding positions. Fig. 17 shows the error of the initial mapping [2] as compared to the mapping after our editing approach. The comparative evaluation shown here relies on the ground truth mapping from SCAPE [3] (Fig. 17a) and TOSCA [7] (Fig. 17b) data sets. We can see that applying our approach improves the mapping in the related regions without causing a degradation of the overall mapping. Another way to measure the quality of a mapping is to morph a mesh into the shape of the other using the mapping. Then, we evaluate the mapping by computing $L_2$ and $L_{inf}$ distortion between the mesh and the morphed mesh to estimate the stretch which occurs in the mapping-based morphing process. Fig. 18 shows the morphing of mesh *A* into mesh *B* using both the initial and new mappings. With our updated mapping (Fig. 18d), the vertices of the head *A* are pulled back to the correct place. This has the advantage of mapping the right density of vertices where needed, which is very important for morphing and in any transfer related to animation attributes (e.g., bones, vertex weights, and blend shapes). Table 2 illustrates an evaluation of the quality of the edited mapping in comparison to the initial mapping. It shows that our edited mapping is as good as or better than the initial mapping when considering the distortion of the morphed mesh. We can see that there is a single case where this measurement of distortion is slightly higher after the map is edited. Even in this case, while the distortion is slightly higher, the edited mapping is clearly superior, as can be seen in Fig. 13.

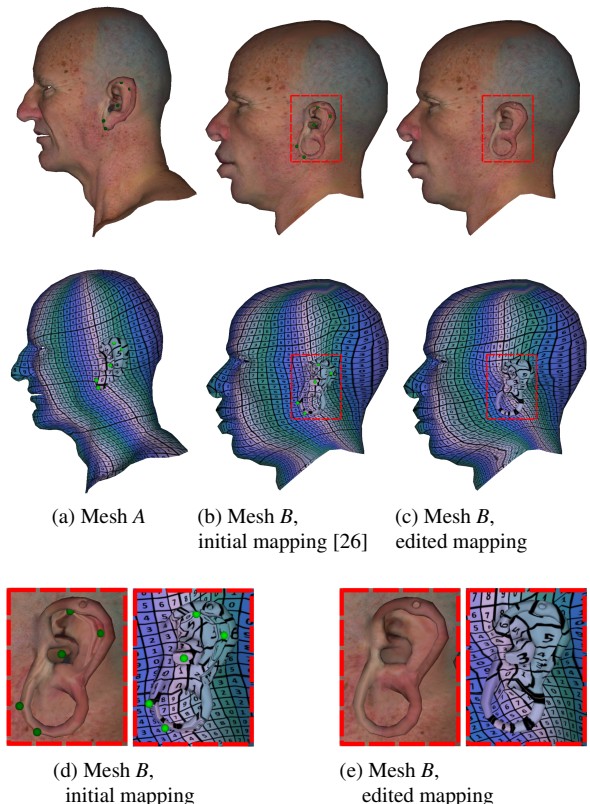

(a) Mesh *A*  (b) Mesh *B*, initial mapping [26]  (c) Mesh *B*, edited mapping

(d) Mesh *B*, initial mapping  (e) Mesh *B*, edited mapping

Figure 16: Map editing in the presence of different genera. Mapping mesh *A* to mesh *B* with different genera leads to distortion in the ear region (outlined in red). Our approach reduces texture transfer issues locally even in regions with higher genera. Apart from the realistic facial textures, we show the lack of distortion and discontinuity through a grid texture.

Table 2: Distortion ratio of the meshes morphed through the edited mapping as compared to the initial mapping. The $L_2$ and $L_{inf}$ ratios correspond to $L_{2_{edited}}/L_{2_{initial}}$ and $L_{inf_{edited}}/L_{inf_{initial}}$, respectively.

| # Figure | 1 | 13 | 15 | 16 | 18 | 19 | 19 | 20 | 20 | 22 | 21 | 21 | 23 | 24 |
|---|---|---|---|---|---|---|---|---|---|---|---|---|---|---|
| $L_2$ ratio | 0.1 | 1.0 | 0.7 | 0.9 | 0.0 | 0.6 | 0.0 | 0.0 | 0.0 | 0.9 | 0.9 | 0.9 | 0.4 | 0.5 |
| $L_{inf}$ ratio | 0.1 | 0.9 | 0.7 | 0.9 | 0.0 | 0.7 | 0.1 | 1.0 | 1.0 | 0.9 | 0.9 | 0.9 | 0.4 | 0.6 |

## 4.3 Comparison

We performed a qualitative comparison of the mapping editing versus the methods of Ezuz and Ben-Chen [9]. We also did comparisons using LIM, SLIM, KM-Newton, and ILSCM to conduct the mapping editing. Furthermore, we compared local editing to global editing using the method of Panozzo et al. [19] and the joint planar method [21].

For the comparison to the method of Ezuz and Ben-Chen [9], we established an initial mapping using a state-of-the-art functional mapping method [17]. Note that we use the raw functional map, without the high-dimensional iterative closest point post-process refinement [18]. Fig. 19 compares the mappings improved using our approach and the method of Ezuz and Ben-Chen [9] (which improves the mapping without any landmark). Note how the added control of the landmarks provides a significantly improved mapping, exactly where intended.

Fig. 21 presents results when LIM, SLIM, KP-Newton, and ILSCM are used to conduct the mapping editing. The comparison through texture transfer visualization shows that ILSCM is superior in adjusting the mapping as compared to LIM, SLIM, and

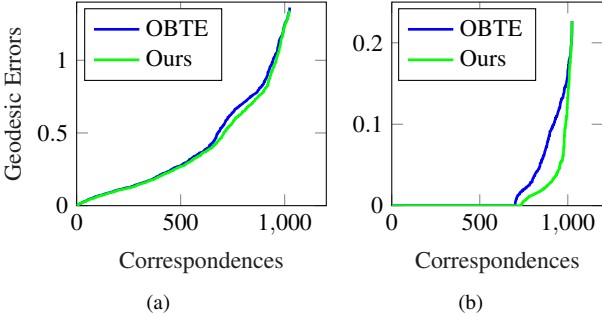

(a)  (b)

Figure 17: Comparison of the initial mapping (OBTE [2]) and our edited mapping. The geodesic errors are calculated for each vertex and sorted in ascending order. (a) and (b) correspond to the examples of Fig. 18 and Fig. 21 (top row), respectively.

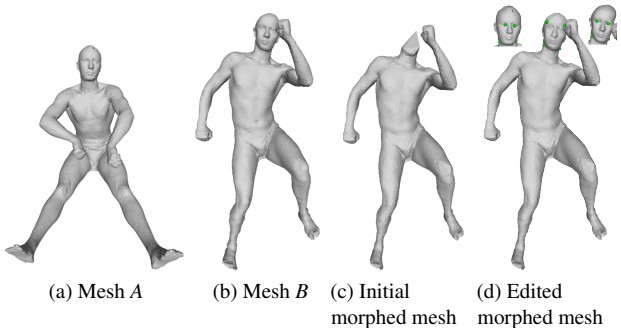

(a) Mesh $A$    (b) Mesh $B$    (c) Initial morphed mesh    (d) Edited morphed mesh

Figure 18: (a)-(b): Initial meshes. (c) With the initial mapping [2], mesh $A$ morphed to mesh $B$ is unable to recreate the head. (d) The two faces on top of the figure show initial meshes and their landmarks used by our approach to improve the mapping. The morphing with the mapping updated by our approach pulls the vertices corresponding to the head to the right place.

KP-Newton. The accompanying video also compares LIM, SLIM, KP-Newton, and ILSCM in editing the mapping for the test case of Fig. 13.

Adjusting the mapping globally requires having the initial constraints, the initial parameters of the method, and the method itself, which is constraining. In addition, some mapping methods, such as that of Nogneng and Ovsjanikov [17], do not let the user guide the process with landmarks, while others, such as OBTE [2], only support a fixed number of landmarks (three or four landmarks for OBTE [2]), which will be insufficient in many cases. Furthermore, we believe that it is advantageous to ensure that changes occur locally, avoiding unexpected changes elsewhere in the mapping. Fig. 20 and 22 compare our approach to two automatic methods. For the automatic methods, the mapping was computed with an initial set of landmarks, and the improved mapping was obtained by adding new landmarks to the initial set. Conversely, our approach only uses the new landmarks to improve the mapping. Fig. 20c (top row) shows that solving for the mapping globally is sometimes as effective as solving it locally. Conversely, Fig. 20c (bottom row) shows that improving the mapping globally introduced artifacts on the fish head as compared to our local refinement (Fig. 20d bottom row), which is exempt of such artifacts away from the edited region. Fig. 22 compares the mappings improved using our approach as compared to solving globally using the WA method of Panozzo et al. [19]. We established an initial mapping using the WA method. Afterwards, with the WA method, we added two additional land-

marks to improve the initial mapping. For our approach, we only consider the two new landmarks in improving the mapping. It can be seen in Fig. 22 that editing the mapping locally was beneficial for this test case as well.

## 4.4 Applications Relying on Surface Mapping

Several applications rely on a mapping between surfaces: texture transfer [29], animation setup transfer [4], and deformation transfer [26]. We use the methods of Sumner et al. [26] and Avril et al. [4] to illustrate how the proposed approach can significantly improve the results of techniques relying on a mapping. Fig. 23 shows a facial transfer result before and after editing. Results demonstrate several issues and unpleasant deformations for fine features, such as strange deformations on the corners of the mouth. With the corrected mapping, these problems disappear. Fig. 24 shows a skeleton transfer [4] result before and after the mapping is edited. Results demonstrate that the joint, that was erroneously positioned outside of the thumb, moves to the right place when improving the surface mapping locally in the thumb region instead of improving the mapping globally over the mesh.

Our approach works even for surfaces with boundaries inside the segments. Such boundaries are commonly encountered with the ears, eyes, nostrils, and mouths of characters. While we constrain the segment boundaries to prevent them from moving, an initial mesh boundary lying inside a segment will be free to move. Leaving these inner boundaries completely free has a negative impact on the deformation. Fig. 25 shows the deformation of the mouth without (c) and with (d) inner boundary fillings. Note here the improvement of the mouth deformation when filling the inner boundary.

## 5 DISCUSSION

Our approach carves a new path in between the more classical shape-preserving methods, which often lose local injectivity, and the more current methods, which formulate the injectivity constraint as part of the optimization. These latter methods typically do not have a bound on the shape-preserving error. In our approach, we are minimizing only the shape-preserving term (i.e., LSCM energy) and iteratively improving the user constraints while maintaining a locally injective map in each iteration (a formal proof is found in Appendix A). We achieve this by carefully controlling the $\lambda$ parameter in Eq. 1. At one extreme, if $\lambda$ is very large (i.e., infinity), the formulation is equivalent to the LSCM formulation. If $\lambda$ is very small, it takes many iterations for the user constraints to be satisfied, or in some cases, the user constraints may ultimately not be satisfied. Our iterative scheme relies on two important observations. If $\lambda$ is 0, the solution is the same as the initial configuration. Therefore, if we start in a locally injective configuration, the final result will be a locally injective configuration. If the initial configuration is locally injective, there always exists a $\lambda$ (however small) that will result in a locally injective configuration, where the user constraints are closer to the target. This scheme will converge to a locally injective configuration. Consequently, we iteratively repeat the optimization to fight against flipped faces, but convergence cannot be guaranteed. It is always possible to design a landmark configuration in which the constraints cannot be met without flipped faces. This is true for the other deformation methods as well. Appendix B demonstrates different failure cases using different deformation methods. In our experiments, except for the examples in Appendix B, the constraints are satisfied (up to numerical precision), even for extreme deformations.

In our results, we improved mappings which were initially computed from a variety of methods [2, 17, 19, 21, 26, 29]. Even if these initial mappings minimize different deformation energies, the fact that we rely on the LSCM conformal energy to edit them did not prevent our approach to improve the mappings. One must keep in mind that the goal of the editing is not to strictly minimize a deformation

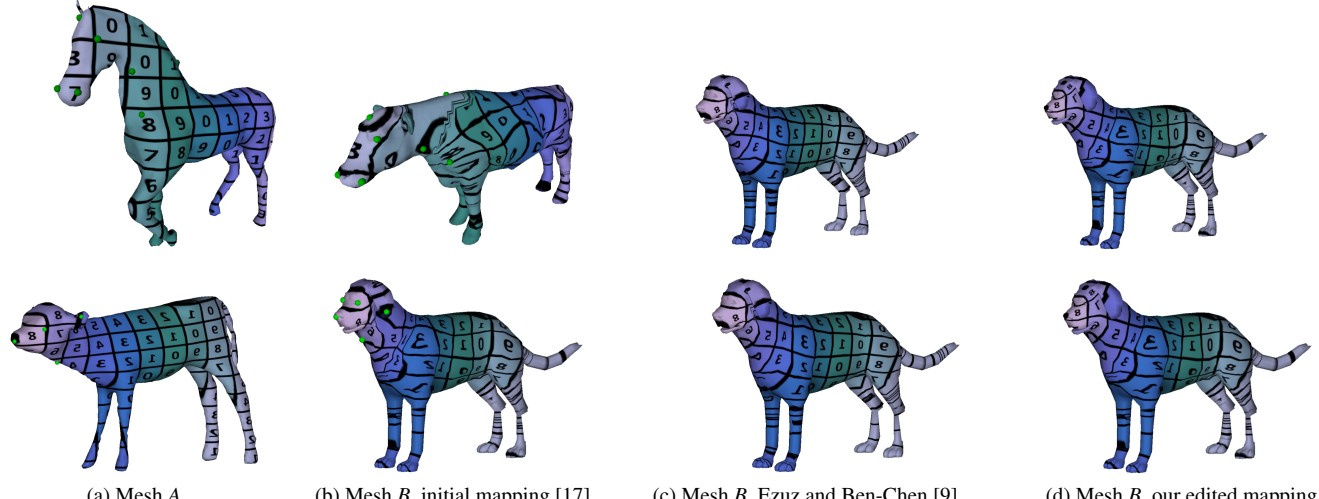

| (a) Mesh *A* | (b) Mesh *B*, initial mapping [17] | (c) Mesh *B*, Ezuz and Ben-Chen [9] | (d) Mesh *B*, our edited mapping |

Figure 19: Qualitative comparison with the deblurring method [9]. Our approach allows more control, and significantly improves the mapping in the neck area.

energy, but to align important semantic features of the objects and maintain injectivity.

We analyzed our results to verify the degree to which the deformation deteriorates the shape of the triangles. We checked 13 of the results found in this paper, and we considered that a detrimental deformation is one in which the angle becomes more than 20 times narrower after deformation. Eleven cases had no such triangles, while the two other cases had two and three, respectively. The worst triangle in our 13 test cases was 24 times narrower than before deformation. Any deformation method is prone to result in thin triangles, so we compared our approach to LIM, SLIM, and KP-Newton for six examples. When looking at the worst triangle found in the deformed meshes, ILSCM performed best for four of the test cases, while KP-Newton performed best for two of the test cases. SLIM and LIM were systematically in third and fourth place behind ILSCM and KP-Newton. Furthermore, our results were better than LIM, SLIM, and KP-Newton in terms of shape preservation and final triangulation, as can be seen in Fig. 12 and in the video. As mentioned earlier, to ensure a fair comparison, we adapted all of the other methods so that they minimize the LSCM energy. For our approach, we minimized the LSCM energy through a least-squares solve. We ran our experiments on a 3.40 GHz Intel Core-i7-4770 CPU with 12 GB of memory, with a MATLAB implementation of our approach. Table 1 shows computation times for the segmentation and the deformation (including mapping extraction) phases. Since our deformation phase is iterative, the time to edit a mapping depends on the size of the mismatching regions and the number of iterations.

## 6 CONCLUSION

We have presented a novel approach for improving surface mappings locally. Our approach is based on a low-distortion region-growing segmentation followed by an independent planar parameterization of each segment. The mapping is then optimized based on an alignment of the user-prescribed landmarks in the parameterization space of each segment. Our joint planar parameterization deformation for the segments is robust, and results in low distortion. Our new iterative LSCM approach can be reused in several contexts where a deformation with low distortion is required. From a practical perspective, our approach has several advantages. It can be used to improve the mapping resulting from any surface mapping method. It also provides a great deal of control, allowing the user to restrict editing to a specific region and to add as few or as many landmarks

as necessary to achieve a desired result.

Our local editing leads to interesting questions which open many avenues for future work. One such prospective area is higher-level landmarks such as lines. This will lead to challenges in terms of easing the interactive placement of these lines on both meshes, but will provide a better set of constraints for the deformation. Another avenue would be to extend the scope to editing deformation transfer. This will combine deformation with editing and enable the user to control animation retargeting.

### ACKNOWLEDGMENTS

This work was supported by Autodesk Inc., Prompt Inc., NSERC, École de technologie supérieure, and the GRAND NCE. We are thankful to Olivier Dionne for his support in the initial phase of the collaboration and Martin de Lasa for his support throughout the collaboration with Autodesk. We want to thank the anonymous reviewers for their valuable comments, as well as Joël Morency and Renée Bourassa for providing 3D characters.

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

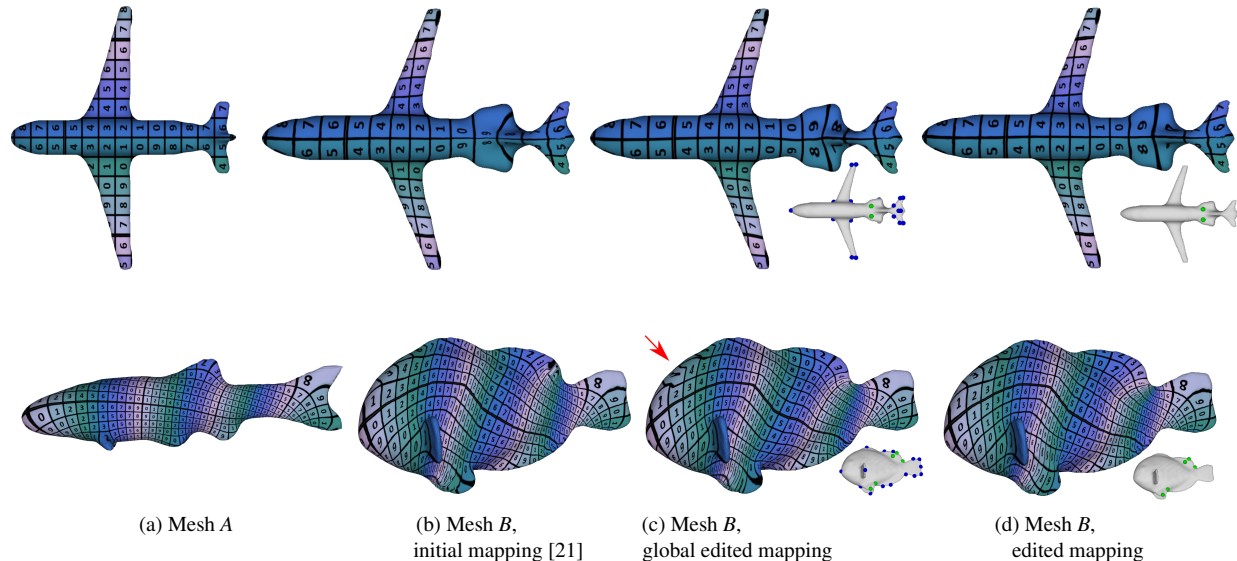

|  (a) Mesh A | (b) Mesh B, initial mapping [21] | (c) Mesh B, global edited mapping | (d) Mesh B, edited mapping |

Figure 20: Qualitative comparison of adjustment of the mapping around the aircraft stabilizer (top row) and the fins (bottom row). (c) Adjustment of the mapping globally using the joint planar method [21]. (d) Adjustment of the mapping locally with our approach. The blue landmarks are used for the initial mapping. Both the blue and green landmarks are used for the global adjustment, while only the green landmarks are required for the local adjustment. The top row shows an example in which the global adjustment of the mapping works well, while for the bottom row, the global adjustment introduces issues in the mapping.

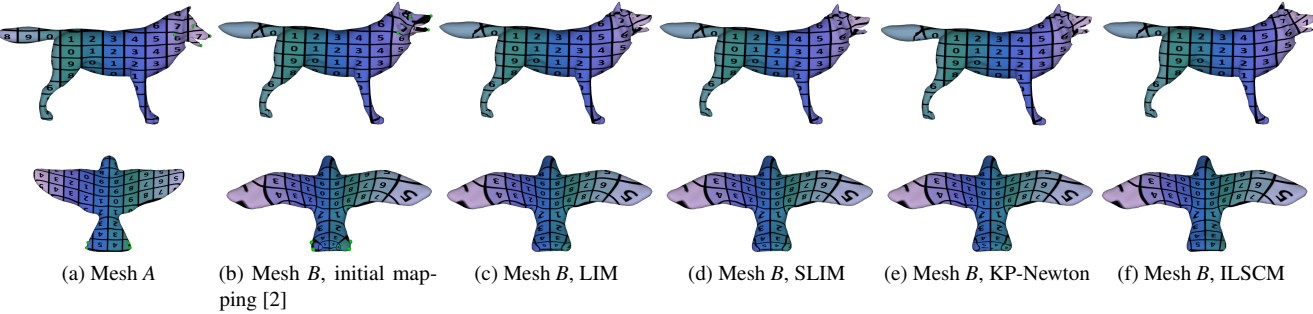

|  (a) Mesh A | (b) Mesh B, initial mapping [2] | (c) Mesh B, LIM | (d) Mesh B, SLIM | (e) Mesh B, KP-Newton | (f) Mesh B, ILSCM |

Figure 21: Qualitative comparison between LIM, SLIM, KP-Newton, and ILSCM to conduct the mapping deformation (minimizing the LSCM energy for all methods). LIM, SLIM, and KP-Newton failed to pull back the wolf muzzle to the correct place. Bottom row shows that ILSCM is superior to LIM, SLIM, and KP-Newton in terms of improving the mapping of the bird tail.

[9] D. Ezuz and M. Ben-Chen. Deblurring and denoising of maps between shapes. *Comput. Graph. Forum*, 36(5):165–174, Aug. 2017. doi: 10.1111/cgf.13254

[10] A. Gehre, M. Bronstein, L. Kobbelt, and J. Solomon. Interactive curve constrained functional maps. *Computer Graphics Forum*, 37:1–12, Aug 2018. doi: 10.1111/cgf.13486

[11] B. Golla, H.-P. Seidel, and R. Chen. Piecewise linear mapping optimization based on the complex view. *Computer Graphics Forum*, 37(7):233–243, 2018.

[12] A. Jacobson, I. Baran, J. Popović, and O. Sorkine. Bounded biharmonic weights for real-time deformation. *ACM Trans. Graph.*, 30(4), July 2011. doi: 10.1145/2010324.1964973

[13] V. G. Kim, Y. Lipman, and T. Funkhouser. Blended intrinsic maps. *ACM Trans. Graph.*, 30(4), July 2011. doi: 10.1145/2010324.1964974

[14] B. Lévy, S. Petitjean, N. Ray, and J. Maillot. Least squares conformal maps for automatic texture atlas generation. *ACM Trans. Graph.*, 21(3):362–371, July 2002. doi: 10.1145/566654.566590

[15] Y. Lipman. Bounded distortion mapping spaces for triangular meshes. *ACM Trans. Graph.*, 31(4), July 2012. doi: 10.1145/2185520.2185604

[16] A. Nguyen, M. Ben-Chen, K. Welnicka, Y. Ye, and L. Guibas. An optimization approach to improving collections of shape maps. *Comput. Graph. Forum*, 30:1481–1491, 08 2011. doi: 10.1111/j.1467-8659.2011.02022.x

[17] D. Nogneng and M. Ovsjanikov. Informative descriptor preservation via commutativity for shape matching. *Comput. Graph. Forum*, 36(2):259–267, May 2017. doi: 10.1111/cgf.13124

[18] M. Ovsjanikov, M. Ben-Chen, J. Solomon, A. Butscher, and L. Guibas. Functional maps: A flexible representation of maps between shapes. *ACM Trans. Graph.*, 31(4), July 2012. doi: 10.1145/2185520.2185526

[19] D. Panozzo, I. Baran, O. Diamanti, and O. Sorkine-Hornung. Weighted averages on surfaces. *ACM Trans. Graph.*, 32(4), July 2013. doi: 10.1145/2461912.2461935

[20] M. Rabinovich, R. Poranne, D. Panozzo, and O. Sorkine-Hornung. Scalable locally injective mappings. *ACM Trans. Graph.*, 36(2), Apr. 2017. doi: 10.1145/2983621

[21] S. Ramachandran, D. Ghafourzadeh, M. de Lasa, T. Popa, and E. Paquette. Joint planar parameterization of segmented parts and cage deformation for dense correspondence. *Computers & Graphics*, 74:202–212, 2018.

[22] P. V. Sander, J. Snyder, S. J. Gortler, and H. Hoppe. Texture mapping progressive meshes. In *Proceedings of the 28th Annual Conference on Computer Graphics and Interactive Techniques*, SIGGRAPH '01, p. 409–416. Association for Computing Machinery, New York, NY, USA, 2001. doi: 10.1145/383259.383307

[23] C. Schüller, L. Kavan, D. Panozzo, and O. Sorkine-Hornung. Locally injective mappings. In *Proceedings of the Eleventh Eurograph-*

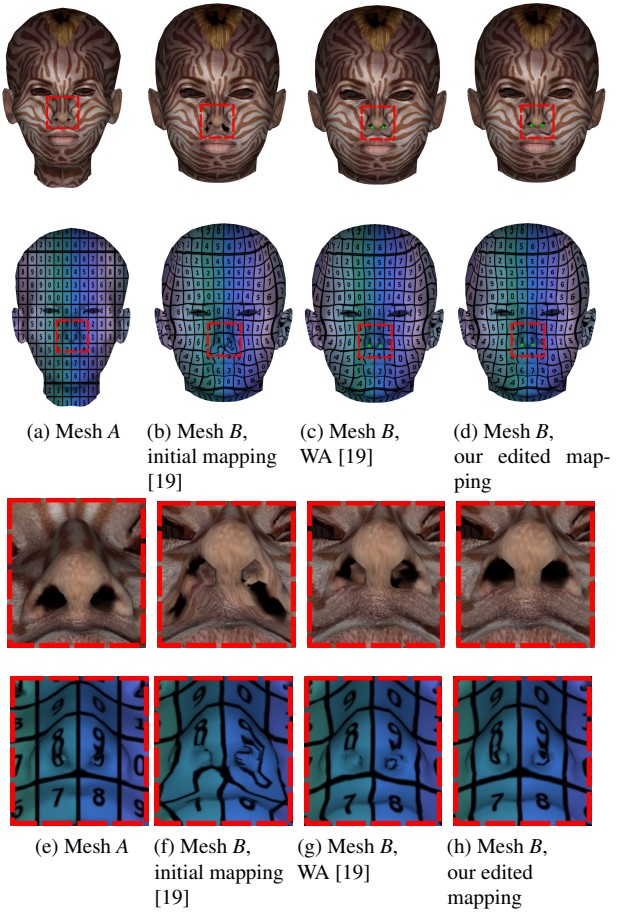

(a) Mesh *A*  (b) Mesh *B*,  (c) Mesh *B*,  (d) Mesh *B*,
          initial mapping    WA [19]       our edited map-
          [19]                             ping

(e) Mesh *A*  (f) Mesh *B*,  (g) Mesh *B*,  (h) Mesh *B*,
          initial mapping    WA [19]       our edited
          [19]                             mapping

Figure 22: Qualitative comparison with the WA method [19] using realistic and grid textures. Our approach significantly improves the mapping in the nostril area.

*ics/ACMSIGGRAPH Symposium on Geometry Processing*, SGP '13, p. 125–135. Eurographics Association, Goslar, DEU, 2013. doi: 10.1111/cgf.12179

[24] A. Sheffer, B. Lévy, M. Mogilnitsky, and A. Bogomyakov. Abf++: Fast and robust angle based flattening. *ACM Transactions on Graphics*, 24(2):311–330, 2005.

[25] J. Smith and S. Schaefer. Bijective parameterization with free boundaries. *ACM Trans. Graph.*, 34(4), July 2015. doi: 10.1145/2766947

[26] R. W. Sumner and J. Popović. Deformation transfer for triangle meshes. *ACM Trans. Graph.*, 23(3):399–405, Aug. 2004. doi: 10.1145/1015706.1015736

[27] R. Veltkamp and F. ter Haar. Shrec 2007-shape retrieval contest, 2007.

[28] M. Vestner, R. Litman, E. Rodolà, A. M. Bronstein, and D. Cremers. Product manifold filter: Non-rigid shape correspondence via kernel density estimation in the product space. *Computer Vision and Pattern Recognition (CVPR)*, Jan 2017.

[29] E. Zell and M. Botsch. Elastiface: Matching and blending textured faces. In *Proceedings of the Symposium on Non-Photorealistic Animation and Rendering*, NPAR '13, pp. 15–24. ACM, New York, NY, USA, 2013. doi: 10.1145/2486042.2486045

## APPENDIX A

To ensure that there is always a single solution, even if $\lambda$ is arbitrarily small, we add a new term $E_{B\star}$ to Eq. 1:

$$E(V) = E_L(V) + E_B(V) + E_{B\star} + E_D(V) \tag{3a}$$

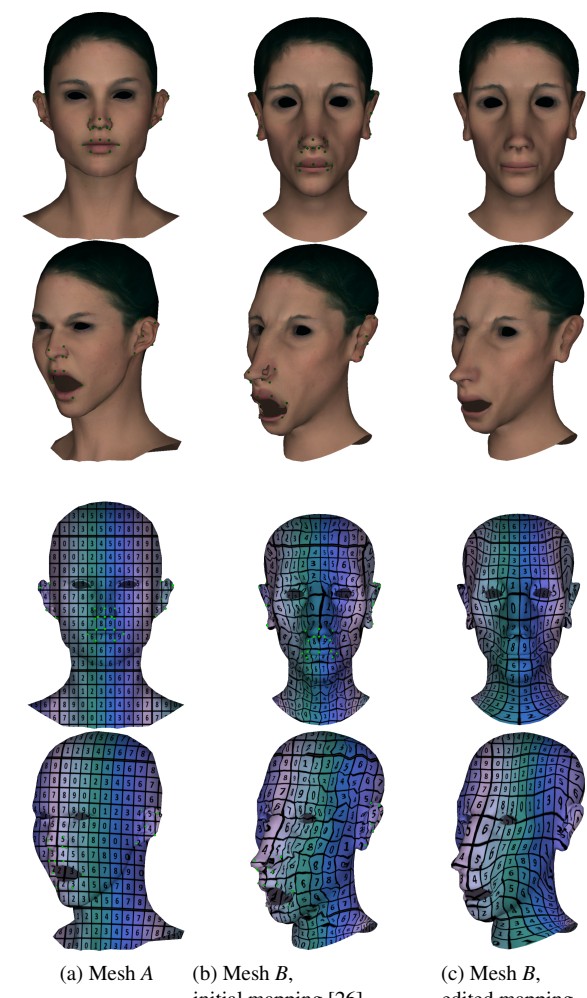

(a) Mesh *A*  (b) Mesh *B*,        (c) Mesh *B*,
          initial mapping [26]    edited mapping

Figure 23: Some mapping issues are not visible on a static mesh with a neutral expression. We show that some subtle mapping issues become obvious and lead to severe problems by opening the mouth. Apart from a realistic skin texture, we also show a grid texture to better show the problems as well as the smoothness of our edited mapping.

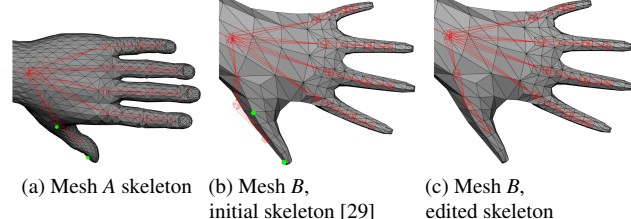

(a) Mesh *A* skeleton  (b) Mesh *B*,           (c) Mesh *B*,
                      initial skeleton [29]   edited skeleton

Figure 24: When using the mapping to retarget attributes, in this case the skeleton, an incorrect mapping will lead to problems, here putting the thumb joint outside of the mesh. By locally editing the mapping, it is easy to fix such issues.

$$E_{B\star}(V) = \xi \sum_{j \in \Omega(\tilde{B}_i)} \left\| v_j - v_j^{old} \right\|^2 \tag{3b}$$

where $v_j^{old}$ denotes the position of vertex $v_j$ at the previous iteration. The energy $E_B$ pulls the vertices of the boundary to where they

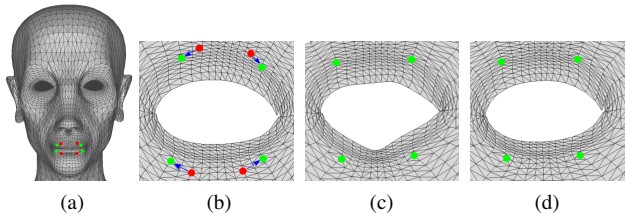

Figure 25: (a) Mesh in 3D. (b) Mesh in 2D. (c) Deformation without filling. (d) Deformation with filling.

correspond given the mapping. The term $E_{B\star}$ pulls the vertices on the boundary of $\widetilde{B}_i$ to their position at the previous iteration. Eq. 3b is weighted by a small constant $\xi = 0.001$ such that in practice the vertices will converge to $map(v_j)$. The previous position $v_j^{old}$ is initialized with the position on the boundary of the ABF of $\widetilde{B}_i$.

Our deformation method proceeds iteratively by finding a sequence of 2D embeddings $V_i$ of a given patch. We show that if the initial embedding of the mesh $V_0$ has no fold-overs, then the resulting embedding at every iteration $V_i$ also has no fold-overs. We prove this by induction. The base case for $i = 0$ is given by the hypothesis, and thus, we are showing that if $V_i$ has no fold-overs, our procedure will yield a configuration $V_{i+1}$ that also has no fold-overs.

At every iteration, the new set of vertex positions $V_{i+1}$ is obtained by solving Eq. 1: $\arg\min_{V_{i+1}}(E(V_i, V_{i+1}, \bar{\lambda})$, where $V_i$ is the embedding in the current iteration, $V_{i+1}$ is the new embedding we are computing, and $\bar{\lambda} > 0$ is a parameter of the algorithm, constant w.r.t. this minimization. We select $\bar{\lambda}$ as follows: we create a monotonically decreasing positive sequence $\lambda_j > 0$ such that $\lim_{j\to\infty}(\lambda_j) = 0$; we solve the optimization problem for the $\lambda_j$ in the sequence and stop at the first element in the sequence $\bar{\lambda} = \lambda_k$ that yields a fold-over free configuration, and we now show that such a $\lambda_k$ always exists.

Let $B(x) \in \mathbb{R}^{n \times n}$, $x \in \mathbb{R}$ and $B_{ij}(x)$ is continuous in $x$ $\forall i, j$.

**Lemma 6.1** $det(B(x))$ is a continuous function in $x$.

*Proof:* We prove by induction on $n$. If $n = 1$, $B(x)$ is a continuous real function. $det(B(x)) = B_{11}(x)$ is also continuous. We assume that the statement is true for $n-1$ and we prove for $n$. We write $det(B(x))$ using the Laplace formula:

$$det(B(x)) = \sum_{i=1}^{n} (-1)^{i+j} \cdot B_{ij}(x) \cdot M_{ij}(x) \qquad (4)$$

where $M_{ij}$ is the minor of the entry $(i, j)$ defined as the determinant of the sub-matrix obtained by removing row $i$ and column $j$ from $B$. As each element of this matrix is also continuous in $x$ and this reduced matrix is $n-1 \times n-1$, it follows from the inductive hypothesis that $M_{ij}$ is also continuous in $x$. As $det(B(x))$ is obtained by using addition and multiplication of continuous functions, it follows that $det(B(x))$ is continuous in $x$. $\square$

**Corollary 6.1.1** *if* $det(B(x)) \neq 0$ $\forall x$ *then* $\frac{1}{det(B(x))}$ *is also continuous in* $x$.

**Lemma 6.2** *if* $det(B(x)) \neq 0$ $\forall x$ *then* $B_{ij}^{-1}(x)$ *is a continuous function in* $x$ $\forall i, j$.

*Proof:* If $det(B(x)) \neq 0$ $\forall x$ then $B(x)$ is invertible $\forall x$ the inverse of a matrix has the following analytic expression:

$$B^{-1}(x) = \frac{1}{det(B(x)))} \cdot C^{\mathsf{T}} \qquad (5)$$

where $C$ is the matrix of co-factors: $C_{ij} = (-1)^{i+j} \cdot M_{ij}$ and $M_{ij}$ is the minor of the entry $(i, j)$. $M_{ij}(x)$ is continuous in $x$ from Lemma 6.1. It follows trivially that $C_{ij}$ is continuous in $x$. Since $det(B(x)) \neq 0$ $\forall x$ it follows from Corollary 1 that $\frac{1}{det(B(x)))}$ is continuous in $x$. Since $B^{-1}(x)$ is obtained by multiplying a scalar function continuous in $x$ by a matrix whose entries are all continuous in $x$, it follows that $B_{ij}^{-1}(x)$ is continuous in $x$ $\forall i, j$. $\square$

$V_{i+1}(\lambda)$ is the minimizer of a quadratic energy function, and therefore, it has the standard least squares analytical solution:

$$V_{i+1}(\lambda) = (A(\lambda)^{\mathsf{T}} \cdot A(\lambda))^{-1} \cdot A(\lambda)^{\mathsf{T}} \cdot b \qquad (6)$$

where the matrix $A$ and vector $b$ are computed from Eq. 1 in a standard way for a least squares solution. The matrix $A(\lambda)$ has the following structure:

$$A(\lambda) = \begin{bmatrix} A_1(\lambda) \\ A_2(\lambda) \\ A_3(\lambda) \\ A_4(\lambda) \end{bmatrix} \qquad (7)$$

where:

1. $A_1$ corresponds to $E_L$ and is a $k \times n$ matrix that encodes the landmark constraints of the patch. These constraints are weighted by $\lambda$ (Eq. 2a).

2. $A_2$ corresponds to $E_B$ and is a $b \times n$ matrix that encodes the boundary constraints of the patch. These constraints are weighted by $\lambda$ (Eq. 2b).

3. $A_3$ corresponds to $E_{B\star}$, and is a $2 \times n$ matrix that constrains two boundary vertices to their positions in the previous iteration. These constraints are weighted by a small constant $\xi$ independent of $\lambda$ (Eq. 2c).

4. $A_4$ corresponds to $E_D$ and is the $m \times n$ matrix from the original LSCM formulation [12], where $n$ is the number of vertices in the patch and $m > n$.

**Lemma 6.3** $A(\lambda)(i, j)$ is a continuous function of $\lambda$ $\forall i, j$.

*Proof:* The matrix $A(\lambda)$ is constructed by stacking four matrices: $A_1$, $A_2$, $A_3$ and $A_4$ shown above. $A_3$ and $A_4$ are independent of $\lambda$ and therefore all their entries are continuous w.r.t. $\lambda$. The entries of $A_1$ and $A_2$ are either 0, and therefore continuous in $\lambda$, or a linear function of $\lambda$, as in Eq. 2a and 2b, and therefore also continuous in $\lambda$. $\square$

**Corollary 6.3.1** $A^{\mathsf{T}}(\lambda)(i, j)$ is a continuous function of $\lambda$ $\forall i, j$.

*Proof:* As the entries of $A^{\mathsf{T}}(\lambda)(i, j)$ are the same as for $A(\lambda)(i, j)$ this follows trivially from Lemma 6.3. $\square$

**Corollary 6.3.2** $(A^{\mathsf{T}}(\lambda) \cdot A(\lambda))(i, j)$ is a continuous function of $\lambda$ $\forall i, j$.

*Proof:* As $(A^{\mathsf{T}}(\lambda) \cdot A(\lambda))(i, j)$ is obtained by multiplying and adding elements of $A$ and $A^{\mathsf{T}}$ that are all continuous in $\lambda$, it follows that $(A^{\mathsf{T}}(\lambda) \cdot A(\lambda))(i, j)$ is a continuous function of $\lambda$ $\forall i, j$. $\square$

**Corollary 6.3.3** $(A^{\mathsf{T}}(\lambda) \cdot b)(i)$ is a continuous function of $\lambda$, $\forall i$.

*Proof:* $b(i)$ is constant in $\lambda$ $\forall i$. $(A^{\mathsf{T}}(\lambda) \cdot b)(i)$ is obtained by multiplying and adding elements of $b$ and $A^{\mathsf{T}}$ that are all continuous w.r.t. $\lambda$, and it therefore follows that $(A^{\mathsf{T}}(\lambda) \cdot b)(i)$ is a continuous function of $\lambda$ $\forall i$. $\square$

**Lemma 6.4** $det(A^{\mathsf{T}}(\lambda) \cdot A(\lambda)) \neq 0$ $\forall \lambda$

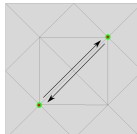

Figure 26: Example of cases where it is impossible to meet the constraints without flipping triangles.

*Proof:* The LSCM paper [12] shows that if we constrain exactly 2 vertices, we obtain a unique solution, which means that:

$$A_{34} = \begin{bmatrix} A_3(\lambda) \\ A_4(\lambda) \end{bmatrix} \tag{8}$$

has rank $n$. Since the matrix $A_{34}$ does not depend on $\lambda$, it follows that $rank(A_{34}) = n \; \forall \lambda$. Since the rank of $A_{34}$ cannot be larger than $n$, and the rank of the resulting matrix does not decrease by adding rows to the matrix, it follows that when stacking $A_1$ and $A_2$ to $A_{34}$ to form the final matrix $A$, $rank(A(\lambda)) = n \; \forall \lambda$. Since the rank of the Gramm matrix is the same as that of the matrix, it follows that $rank(A^\mathsf{T}(\lambda) \cdot A(\lambda)) = n \; \forall \lambda$. Since $A^\mathsf{T}(\lambda) \cdot A(\lambda)$ is a $n \times n$ matrix, this means that $A$ has full rank, therefore $det(A(\lambda)) \neq 0 \; \forall \lambda$. □

**Lemma 6.5** $V_{i+1}(\lambda)$ *is continuous in* $\lambda$.

*Proof:* $V_{i+1}(\lambda) = (A(\lambda)^\mathsf{T} \cdot A(\lambda))^{-1} \cdot A(\lambda)^\mathsf{T} \cdot b$. From Lemmas 6.2, 6.3.2, and 6.4, it follows that $(A(\lambda)^\mathsf{T} \cdot A(\lambda))^{-1}(i,j)$ is continuous in $\lambda \; \forall i, j$ and from Corollary 6.3.3 $(A^\mathsf{T}(\lambda) \cdot b)(i)$ is a continuous function of $\lambda \; \forall i$. As each element of $V_{i+1}(\lambda)$ is computed as a sum or product of functions continuous in $\lambda$, it follows that $V_{i+1}(\lambda)$ is continuous in $\lambda$. □

**Lemma 6.6** *if* $\lambda = 0$, *then* $V_{i+1} = V_i$

*Proof:* $\lambda = 0$ reduces the linear system to only $A_3$ and $A_4$. □

**Theorem 6.7** $\exists k > 0 \; s.t. \; V_{i+1}(\lambda_k)$ *has no fold-overs.*

*Proof:* From Lemma 6.6, if $\lambda = 0$, then $V_{i+1}$ has no fold-overs. Since $V_{i+1}$ is continuous in $\lambda$ (Lemma 6.5), it follows that for all vertex positions $\exists \hat{\lambda} > 0 \; s.t. \; \forall \lambda, 0 < \lambda < \hat{\lambda}, V_{i+1}(\lambda)$ has no fold-overs. Since the sequence $\lambda_j$ is monotonically decreasing and $\lim_{j \to \infty}(\lambda_j) = 0$, it follows that $\exists k \; s.t. 0 < \lambda_k < \hat{\lambda}$. It follows that $V_{i+1}(\lambda_k)$ has no fold-overs. □

By proving Theorem 6.7, we show that at every iteration, our embedding $V_{i+1}$ has no fold-overs and thus yields an injective map.

## APPENDIX B

Our deformation method guarantees progress toward meeting the landmark constraints while being free of flipped faces, but it cannot guarantee that the user constraints will be satisfied. In fact, there are cases where it is impossible to meet these constraints, such as the example in Fig. 26. There are also "hard" cases (Fig. 27) where, while it might be possible to find a deformation that meets the constraints, deformation methods, such as LIM, SLIM, KP-Newton and our approach, are not able to find it.

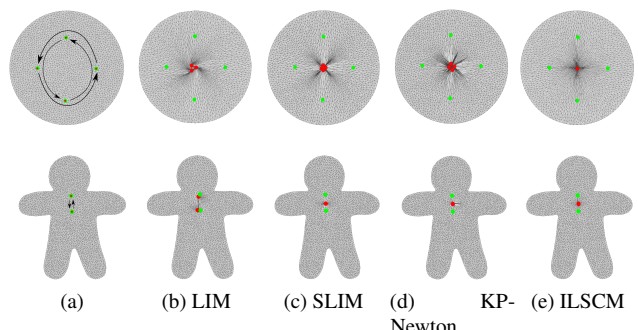

|     |         |          |      |        |
|-----|---------|----------|------|--------|
| (a) | (b) LIM | (c) SLIM | (d) KP-Newton | (e) ILSCM |

Figure 27: With the design of these landmark configurations, moving the landmarks (red dots) to their final positions (green dots) makes it hard to converge without any flipped faces. LIM, SLIM, KP-Newton, and ILSCM are unable to simultaneously meet the landmark constraints and avoid having flipped triangles.