# OpenReview forum: "Local Editing of Cross-Surface Mappings with Iterative Least Squares Conformal Maps"
_graphicsinterface.org/Graphics_Interface/2020/Conference — GI 2020_

### Official Review · AnonReviewer1 · 2020-01-08
**Neat idea of locally refining dense surface-to-surface maps, but lacking quality evaluation of robustness**

**Confidence:** 4
**Rating:** 5

**Review:**

This paper presents a technique to incrementally improve dense surface-to-surface maps by minimizing conformal distortion in local parameterized patches and ensuring injectivity during optimization. Unlike previous work that updates the entire surface correspondence on insertion of landmarks, the key emphasis of this work is to only edit the map locally. The following can be viewed as contributions of the paper:

(1) The idea of locally editing the surface-to-surface map via construction of local cross parameterizations.

(2) A different approach to planar deformation that minimizes conformal distortion and preserves local injectivity.

While the techniques proposed in this paper might be useful in parts for a surface map editing tool, I think the paper has a number of weaknesses. In particular:

(1) The authors should clarify what energy they minimize with SLIM/LIM - symmetric Dirichlet or conformal AMIPS? If the former is used, then it is not fair to claim that SLIM & LIM demonstrate larger overall distortion compared to ILSCM when a conformal distortion metric is used for comparison. It is expected that SLIM & LIM will demonstrate larger angle distortion than LSCM inspired approaches if they minimize an isometric energy.

(2) The authors should provide timing comparisons against related work. While the primary cost of the method appears to be a numerical factorization for every iteration of the deformation scheme on each patch, not much can be said (or is said) about the convergence behavior (such as number of iterations) of ILSCM. However, the lack of theoretical guarantees can be compensated with  performance benchmarks of just the segmentation and deformation schemes against related work on a large dataset of meshes (e.g., http://staff.ustc.edu.cn/~fuxm/projects/ProgressivePara/dataset.html).

(3) Most of the textures used in the paper are fairly smooth, and as a result, its difficult to ascertain whether the resulting mappings are smooth, for example, near the patch boundaries. Figure 14 by itself is not very particularly convincing of the robustness of the approach.

(4) ABF++ is used to find the initial parameterization of the patches, but it is not guaranteed to be injective. Presumably, other methods such as SLIM with conformal AMIPS can be used to get an initial injective mapping.

Overall, I can see parts of this framework being used in a production pipeline. However, in my opinion a better evaluation is required to make up for the lack of theoretical guarantees (in particular, via testing the performance characteristics of the deformation scheme and the quality of the final mapping in terms of distortion on a larger dataset, rather than just the 13 meshes in the paper).

---

### Official Review · AnonReviewer3 · 2020-01-09
**Localized Improvement of Conformal Distortion is An Interestind Idea**

**Confidence:** 3
**Rating:** 6

**Review:**

This paper presents a method to improve upon existing maps between surfaces by a supervised method: the user provide landmarks of places where they deem to have a bad map, which are supposedly localized. The method then automatically devises a local region around these benchmarks, and optimizes for an LCSM-like energy where the landmarks are iteratively moved (with slow enough steps) to the desired positions.

This is a nice idea that I think is worth publishing, and the method is decently---though minimally---evaluated and compared. i am convinced it is generally a good idea to work this way. However, I have a few reservations about the method and the evaluation that should be resolve in a revision:

1. Did I miss where the bound on the geodesic distance is? I mean for the computation of the size of the patch. Also, why is geodesic a good idea? it would seem that one would like to grow the patch until they reach boundaries that are already good enough.

2. What is the distortion measured in Figures 5 and 6 exactly?

3. There are definite cases where the user can make things worse but "ruining" the map, albeit not making the constraints impossible, which is mentioned in Appendix B. how would you converge then?

4. To be a fair comparison, since the method is supervised, one would have to compare the automatic method with the full set of landmarks obtained (both original and new input). Is this the case?

5. I don't understand Figure 18: why is the deformation in (c) not interpolating the green landmarks?

6. Sometimes it's hard to see the difference in the maps, for instance in Figure 2. The authors say that visualization is a good way to see conformal distortion, but a scalar color map of the local distortion in some cases would help much to see the contribution.

---

### Official Review · AnonReviewer2 · 2020-01-09
**Marginal contribution, but perhaps good enough**

**Confidence:** 4
**Rating:** 6

**Review:**

The paper presents a new method to incorporate user constrains in improving cross-surface mappings. After growing regions around user point constraints and parameterizing them on both involved shapes into 2D, they perform a 2D optimization to align the boundaries and satisfy the points constraints while keeping the distortion low. As distortion, they chose LSCM energy. In order to prevent flips, they use an iterative energy where once flips are detected, they lower the weight of the soft constraints.

Clarity:
The paper is written well and is easy to read.
A few things: in 3.2, "we fill any artificial internal boundaries" --  unclear at this point. It's explainer further in the text, but the intuition should be as soon as those are introduced.

Missing Related Work:
[A] Jason Smith and Scott Schaefer. 2015. Bijective parameterization with free boundaries. ACM Trans. Graph. 34, 4, Article 70 (July 2015)

Notes and Questions to be addressed in the final publication:

As far as I understand, the contribution of SLIM [15] is not to introduce energies that go to infinity once a triangle degenerates, those were introduced before ([A], [Schuller  et al. 2013; Fu et al. 2015]). Rather, they came up with an strategy how to minimize such energies using the local-global method.

I'm not exactly clear how the initial regions are selected in 3.1. Is the method robust to user error here, as long as the different sets of constraints are in different regions? If so, why not use a clustering mechanism?

"LIM and SLIM often produce ..." -- needs a reference to the comparison figure(s).

In 3.2: "... and around each corresponding position $cp_{B->A}(i)_j$." -- so if the point is inside the triangle, all the adjacent triangles are taken?

"One disadvantage ... Is that it can contain concavities" -- I'm not sure what's the issue with concavities. I would understand if the authors were using Tutte's embedding, but it's ABF++, so it's unclear.

Have the authors considered an iterative region growing strategy instead of their potentially unstable heuristic 3.2 with a fixed threshold of twice the geodesic distance? One can imagine, e.g. Parameterizing the patch, measuring the distortion, adjusting the patch.

In 3.3.1, I'm not totally sure which energies were tested in SLIM. Exponential Dirichlet via their quadratic proxy? Symmetric Dirichlet? In any case, they don't guarantee injectivity because of their choice of energy, it's the line search strategy that they use. Which brings me to the question: instead of adjusting the \lambda weight in the optimization, have the authors considered, say, a first-order optimization method (e.g. Gradient descent/projected gradient descent/augm lagrangian, etc.), with the line search constraints like in [A]?

Some implementation details are missing: how is LSCM energy minimized? CG, like in the original paper?

Fig 16. Is not great, the texture is too smooth and even to see issues clearly. Maybe replace with the number grid?

Typos:
Panozzoet al. -> Panozzo et al.

Conclusion:
I think it's not a large, but a contribution worthy of publication. Before publication, however, the aforementioned issues and questions should be addressed in the text.

---

### Meta-Review · Area_Chair1 · 2020-01-09

**Recommendation:** Accept
**Confidence:** 3

**Metareview:**

The three reviewers found that the paper makes a small contribution within the domain on cross-surface mapping, however the paper would have benefited from deeper evaluation / verification. There are also a number of questions brought up by the reviewers that should be addressed before publication (e.g., how initial regions are selected, where the bound on the geodesic distance is, if an iterative growing region strategy was considered, the energies tested in SLIM, if the resulting mappings are smooth near patch boundaries, and so on). In light of the review scores and concerns, it appears that this paper is slightly above the bar and may be acceptable for publication.

---

### Decision · Program_Chairs · 2020-01-11

Accept